# Vitamin D in Systemic Sclerosis: A Review

**DOI:** 10.3390/nu14193908

**Published:** 2022-09-21

**Authors:** Mattia Perazzi, Enrico Gallina, Giulia Francesca Manfredi, Filippo Patrucco, Antonio Acquaviva, Donato Colangelo, Mario Pirisi, Mattia Bellan

**Affiliations:** 1Department of Translational Medicine (DIMET), Università del Piemonte Orientale UPO, Via Solaroli 17, 28100 Novara, Italy; 2Department of Internal Medicine, Rheumatology Unit, “AOU Maggiore della Carità”, 28100 Novara, Italy; 3Department of Health Sciences (DSS), Università del Piemonte Orientale UPO, 28100 Novara, Italy; 4Center for Translational Research on Autoimmune and Allergic Disease (CAAD), Università del Piemonte Orientale UPO, 28100 Novara, Italy

**Keywords:** systemic sclerosis, vitamin D, cholecalciferol

## Abstract

(1) Background: In the present paper we aimed to review the evidence about the potential implication of vitamin D in the pathogenesis and management of systemic sclerosis (SSc); (2) Methods: we performed a review of the literature looking for studies evaluating the potential role of vitamin D and its analogs in SSc. We searched the PubMed, Medline, Embase, and Cochrane libraries using the following strings: (vitamin D OR cholecalciferol) AND (systemic sclerosis OR scleroderma). We included cohort studies, case-control studies, randomized controlled trials, and observational studies. (3) Results: we identified nine pre-clinical and 21 clinical studies. Pre-clinical data suggest that vitamin D and its analogs may suppress fibrogenesis. Clinical data are concordant in reporting a high prevalence of hypovitaminosis D and osteoporosis in SSc patients; data about the association with clinical manifestations and phenotypes of SSc are, conversely, far less consistent; (4) Conclusions: in vitro data suggest that vitamin D may play an antifibrotic role in SSc, but clinical data confirming this finding are currently lacking. Hypovitaminosis D is common among SSc patients and should be treated to reduce the risk of osteoporosis.

## 1. Introduction

Vitamin D_3_, also known as cholecalciferol, is an inactive fat-soluble hormone precursor produced in the skin as a result of ultraviolet activation of 7-dehydrocholesterol, but also ingested, in a small amount, from food. Its active form is 1,25-dihydroxycholecalciferol (1,25(OH)_2_D_3_) or calcitriol, which plays a paramount role in calcium and phosphate homeostasis by activation of calcium absorption in the small intestine and the stimulation of osteoclastic maturation [1,2].

The activation process of cholecalciferol consists of two sequential hydroxylations, the first occurring in the liver, and the second, mediated by CYP27B1 (also called 1α-hydroxylase), taking place in the kidney [3]. Immune cells can express CYP27B1, allowing paracrine and autocrine activity in addition to the endocrine activity of 1,25(OH)_2_D_3_ [3,4,5]; however, while CYP27B1 expression is upregulated by the parathyroid hormone (PTH) and downregulated by calcium concentration in the kidney, its expression in immune cells is independent of endocrine modulation [5,6,7,8].

The pleiotropic effects of calcitriol are mediated through the vitamin D receptor (VDR), a nuclear receptor that heterodimerizes with retinoid X receptor (RXR), and translocates to the nucleus, where it binds Vitamin D response elements (VDREs) in the promoters of Vitamin D-responsive genes [9,10]. VDR is almost ubiquitous, which explains in part the complexity of 1,25(OH)_2_D_3_ activity, going well beyond calcium and phosphate metabolism [11]. Indeed, Vitamin D regulates innate and adaptive immunity by the modulation of the production of pro-inflammatory and anti-inflammatory cytokines [10,12,13,14,15], the suppression of T cells proliferation [10,16], the induction of the shift from a T-helper (Th)1 and Th17 to Th2 phenotype [14,15,17]. Vitamin D also regulates myeloid differentiation towards monocytes and granulocytes while inhibiting the differentiation towards dendritic cells; by doing it, vitamin D reduces the inflammatory activity of antigen-presenting cells (APC), counteracting antigen presentation and the production of cathelicidin antimicrobial peptide (CAMP) and defensin β2 [16,18,19]. Vitamin D also acts on endothelial proliferation, stimulating angiogenesis [20]. Indeed, historically, before antibiotic discovery, vitamin D was used as an antitubercular treatment, due to the enhancement of innate immunity against mycobacterium rather than to a direct antitubercular activity [21,22].

In consideration of its immunoregulatory properties, Vitamin D has been studied over the past years to evaluate its use in the modulation of immune system activity in clinical practice [9,23,24,25]. Specifically referring to systemic sclerosis (SSc), Vitamin D has regulatory activities on different pathogenetic mechanisms of this rare condition: immunity, peripheral vasculopathy, and fibrosis [26]. This is why its application in this context has been explored in the last years; in this study, we aim to review the current evidence about the potential implication of Vitamin D in the pathogenesis and treatment of systemic sclerosis (SSc), starting from pre-clinical data and then moving to the clinical setting.

## 2. Materials and Methods

This review was conducted following the requirements of the Preferred Reporting Items for Systematic Reviews and Meta-Analysis (PRISMA). The data extraction and data synthesis were performed independently by the authors (M.P.), (E.G.) and (M.B.) based on the PICO components (Population; Intervention; Comparison; Outcome). We performed a review of the literature, looking for studies evaluating the potential role of vitamin D and its analogs in SSc. On 18 March 2022, we searched the following databases: PubMed, Medline, Embase, and Cochrane library. We used the following strings: (vitamin D OR cholecalciferol) AND (systemic sclerosis OR scleroderma). We restricted the search to the last 10 years and we included papers that fulfilled the following inclusion criteria:-The availability of the full version of the paper online;-English language;-Study design: either cohort study (prospective or retrospective), case-control study, randomized controlled trial, or observational study;-Studies addressing preclinical and clinical effects of vitamin D in the context of SSc.

Therefore, we excluded letters, abstracts, conference abstracts, comments, case reports, reviews, papers with no English version available, and studies not directly assessing the role of vitamin D in SSc.

Based on inclusion and exclusion criteria, we were able to retrieve 605 papers. After careful, independent screening by two investigators, we finally selected 30 studies, as shown in Figure 1.

## 3. Results

### 3.1. Preclinical and Experimental Studies

Among the 30 selected studies, nine reported pre-clinical findings on mouse models and fibroblast cultures. We summarized the main laboratory findings in Table 1.

Taking together the current body of evidence derived from in vitro studies, consistent data support the hypothesis that vitamin D can modify the fibrogenic activity of fibroblasts, suppressing the pro-fibrotic tissue growth factor (TGF)-β/small mother against decapentaplegic (SMAD) signalling [27,29]; paricalcitol, a noncalcemic analog of vitamin D, was shown to suppress the expression of periostin and collagen 1A1 in dermal fibroblasts exposed to TGF-β and Th2 cytokines [30]. Similarly, 17,20S(OH)_2_pD and calcitriol induce the production of metalloproteinases, such as MMP-1 and BMP-7, involved in matrix degradation [35].

Consistent with these data, the anti-fibrogenic activity of vitamin D has been reported in murine models of scleroderma. Specifically, in mice with bleomycin-induced skin fibrosis, the treatment with systemic topical vitamin D analogs reduced the amount of fibrosis [27,28,29,30] by down-regulating TGF-β signalling and by modulating cytokine mediators, such as IL-13, TNF-α, IL-6, IL-17, and IL-12p70 [34]. These findings suggest that the effect of vitamin D in this context is in part directly related to its activity on crucial pathways of fibrosis development and in part to the modulation of the immune system; indeed, in blood samples collected from SSc patients, vitamin D supplementation increased IL-10 production by T-reg lymphocytes providing a “suppressive” cytokine milieu able to modulate immune response [31].

The activity of vitamin D on fibrogenesis is VDR-dependent; indeed, in VDR knockout mice, skin fibrosis is enhanced after exposure to pro-fibrotic stimuli [33]. Moreover, VDR expression is decreased in fibroblasts of SSc patients and murine models of SSc and VDR downregulates TGF-β/SMAD signalling [29].

### 3.2. Clinical Studies

In Table 2 we report the clinical studies included in the present review. We identified a total of 21 papers; out of them, 15 were case-control studies, three were retrospective cohort studies, and three were cross-sectional studies.

Looking at the current literature, there is a concordance about the high prevalence of hypovitaminosis D among SSc patients [43,56]. When compared to healthy controls, patients affected by SSc showed generally lower vitamin D plasma concentrations, even in the case of cholecalciferol supplementation [38,39,42,47,48,49,50,51,54,55]. Interestingly, when the levels of 25(OH)D were compared among patients with different disease phenotypes, the results were controversial. Indeed, Corrado et al. [39] found that, in patients with the diffuse cutaneous form (dcSSc), 25(OH)D levels were significantly lower than in limited cutaneous form (lcSSc); moreover, vitamin D levels were inversely correlated to Rodnan skin score in other papers [42,51]. However, other authors did not confirm these findings [43,44,50], reporting that the extent of skin involvement was not associated with vitamin D levels in SSc. This discrepancy is also evident in the relationship between vitamin D concentration and autoantibody profile: anti-Scl70 positivity was associated to lower vitamin D levels by some authors [37,40], while others failed to disclose this association [43,44,49]. Interestingly, Giuggioli et al. reported that SSc patients who undergo vitamin D supplementation show higher vitamin D plasma concentrations and are more likely to be anticentromere positive [45].

As expected, Vitamin D plasma concentration is inversely correlated with bone mass density in SSc patients [37,38,40] and, therefore, vitamin D deficiency is also paralleled by an impairment of bone mineral density (BMD), which is significantly lower in SSc patients than in healthy controls [37,38,39,54]. According to Rios-Fernandez et al., the prevalence of osteopenia/osteoporosis in a cohort of SSC patients was 77%, higher than that observed in an age- and sex-matched control group (50%; *p* < 0.0001) [36].

The investigations assessing the possible association of vitamin D levels with specific SSc clinical manifestations led to conflicting results; while some authors failed to disclose any association [39,47], others showed a weak association with some specific domain of disease. In particular, Caimmi et al. [53] and Park et al. [46] reported that vitamin D deficiency is an independent risk factor for the development of digital ulcers. Moreover, some reports suggest a potential association with the degree of lung involvement, since lower vitamin D levels were reported in patients with bibasal interstitial fibrotic changes in the lung [44], with vitamin D levels bearing a weak direct association with diffusing lung capacity of the lung for carbon monoxide (DLCO), according to other papers [36,43].

Finally, two papers assessed the potential additional risk of developing SSc in patients with specific VDR polymorphisms. The results are, once more, controversial. Indeed, while Kamal et al. [41] found that the ApaI and TaqI polymorphisms did not significantly affect SSc susceptibility, according to Li et al. [52], ApaI and BglI polymorphism genotypes were significantly associated with the risk of SSc in a case-control study on 100 SSc patients and 100 healthy controls.

## 4. Discussion

Vitamin D is a pleiotropic molecule that became the subject of intense scrutiny in the last decades because of its novel and putative activities in previously unexpected fields of human physiology and disease. A hot topic is the possible implication of vitamin D in the pathogenesis of autoimmune diseases [57,58]; indeed, vitamin D has a wide and well-characterized activity on the immune system in vitro, the real relevance of which, in vivo, is highly debated. In the present paper, we reviewed the current literature about vitamin D and SSc.

According to our review, pre-clinical data would strongly support the potential use of vitamin D and its non-calcemic analogs in the treatment of fibrosis in SSc. Indeed, different compounds have been used in the past, both in vitro and in murine models, showing an anti-fibrotic effect. This activity is VDR-related and is associated with the suppression of fibrogenic pathways and the modulation of the immune system. However, these findings are not supported by in vivo data, since there is a lack of trials specifically assessing the impact of vitamin D supplementation on SSc-related endpoints. Vitamin D use for the management of autoimmune diseases has been explored in the past, particularly in the context of inflammatory arthritis and systemic lupus erythematosus, with conflicting results [59,60,61]. More generally, the results obtained in vivo are far less promising than the ones suggested by in vitro or murine models. Indeed, differences in the concentration and the type of compounds used may account for many of these discrepancies; moreover, we should consider that in vitro models are very simple and far from being representative of the high complexity of a biological system. Taking all of these considerations into account, although the current body of evidence supports the need for clinical trials assessing the effectiveness of vitamin D supplementation on SSc, the chance of demonstrating real effectiveness is not high.

What is instead well described in vivo is the very high proportion of SSc patients showing inappropriately low levels of vitamin D. Hypovitaminosis D is a common issue in rheumatic patients [62,63,64]; although vitamin D deficiency is highly prevalent in the general population, rheumatic patients are at even higher risk, for several reasons: the chronic use of drugs affecting vitamin D metabolism, a reduced sun exposure, and inappropriate food intake or malabsorption. Malabsorption might also affect the metabolism of other fat soluble vitamins, such as vitamin A, K and E, although, to the best of our knowledge, no data are available in the literature in the specific context of SSc patients. However, it must be noted that the first step of vitamin D metabolism takes place in the skin, which is almost invariably involved along the course of SSc. The fibrotic changes observed in SSc patients, along with reduced sun exposure, may justify the increased risk of hypovitaminosis observed in the disease; in support of this hypothesis is the fact that some authors reported an inverse correlation between skin involvement and vitamin D levels [42,51]. Moreover, lower vitamin D levels have been observed in patients showing anti-topoisomerase positivity, which is generally associated with the diffuse variant of SSc. It should be acknowledged, however, that these observations have not been confirmed by other authors and would be better investigated in appropriately designed cohort studies.

Certainly, the huge prevalence of hypovitaminosis D puts SSc patients at a particularly high risk for osteoporosis; it is well known that osteoporosis is a common comorbidity of rheumatic conditions [65,66]. Many different factors contribute to bone loss in rheumatic conditions: first of all, chronic inflammation has an impact on the pathways involved in the regulation of the physiological bone turnover, but also immobility, vitamin D deficiency, and the chronic use of drugs such as glucocorticoids, with a detrimental effect on bone health have a major role in the pathogenesis of osteoporosis. Looking at SSc patients in particular, it should be recognized that osteoporosis is quite common, with an estimated prevalence ranging from 6.7 to 51.1% [67]; such a high prevalence accounts for the increased risk of osteoporotic fractures, which has been described in SSc [68]. In a Taiwanese cohort, out of 1712 SSc patients, 54 patients developed vertebral fractures, 17 developed hip fractures, and seven developed radius fractures along a median follow-up of approximately five years. The incidence rate ratios were increased in comparison to a group of controls; older age, female gender, using daily prednisolone equivalent to >7.5 mg, and bowel dysmotility treated with intravenous metoclopramide were all risk factors for osteoporotic fractures [69]. These findings reinforce the need for a systematic assessment of bone health in SSc patients with the implementation of all those interventions required to prevent bone loss and osteoporotic fractures, among which are the correction of vitamin D status. Whether SSc patients may require a specific vitamin D supplementation regimen has not been evaluated before. It has been previously reported that patients with autoimmune/inflammatory conditions may show an impairment of the vitamin D/PTH axis possibly related to the chronic inflammatory state [58,70]; this aspect and the possibly defective cutaneous activation of cholecalciferol suggest that SSc patients may need a higher dosage of vitamin D to correct cholecalciferol deficiency. This is still an open issue that should be elucidated with appropriately designed clinical trials.

We also examined data about the potential association of vitamin D levels with specific disease domains; this is probably the most controversial aspect, with the highest heterogeneity in the current literature. Data suggest that vitamin D may be protective against the development of digital ulcers and that patients with decreasing vitamin D concentrations over time are at higher risk for this complication, thus low vitamin D levels seem to play a role in risk factors rather than being a consequence of SSc disease activity [53]. This is possibly related to a direct beneficial effect of vitamin D on microcirculation, which has been demonstrated previously in healthy subjects [71,72]. The data on lung involvement are somehow conflicting; for example, Trombetta et al. reported an association between lower vitamin D levels and severe lung involvement at the chest CT scan, while vitamin D levels did not correlate with DLCO, as conversely reported by Rios-Fernandez and Groseanu, with weak associations. Vitamin D has already been associated with lung fibrosis in other clinical settings, and this led to the postulation of a potential role for this hormone in the management of interstitial lung diseases [73]. To further support this hypothesis, vitamin D was reported to mitigate the development of lung fibrosis in a well-described model of idiopathic pulmonary fibrosis and interstitial lung disease: bleomycin-induced lung fibrosis [74]. Moreover, vitamin D deficiency exacerbates the development of lung fibrosis in this murine model through the overactivation of TGF-β/Smad signalling, the same fibrogenic pathway which is suppressed by vitamin D activity in dermal fibroblasts of SSc patients [75]. Taking these findings together, low vitamin D levels might represent a potential risk factor for ILD development. Once more, this preclinical observation may suggest a beneficial effect of cholecalciferol supplementation in SSc patients which goes beyond the skeletal effect of this hormone, although specific interventional studies should assess this topic.

## 5. Conclusions

While in vitro data suggest that vitamin D may play an antifibrotic role which may be promising in the management of SSc, clinical data confirming this finding in vivo are currently lacking. However, vitamin D deficiency is particularly common in SSc patients, and its status should be carefully assessed and corrected in order to reduce the risk of osteoporosis and fractures, which are high among SSc patients.

## Figures and Tables

**Figure 1 nutrients-14-03908-f001:**
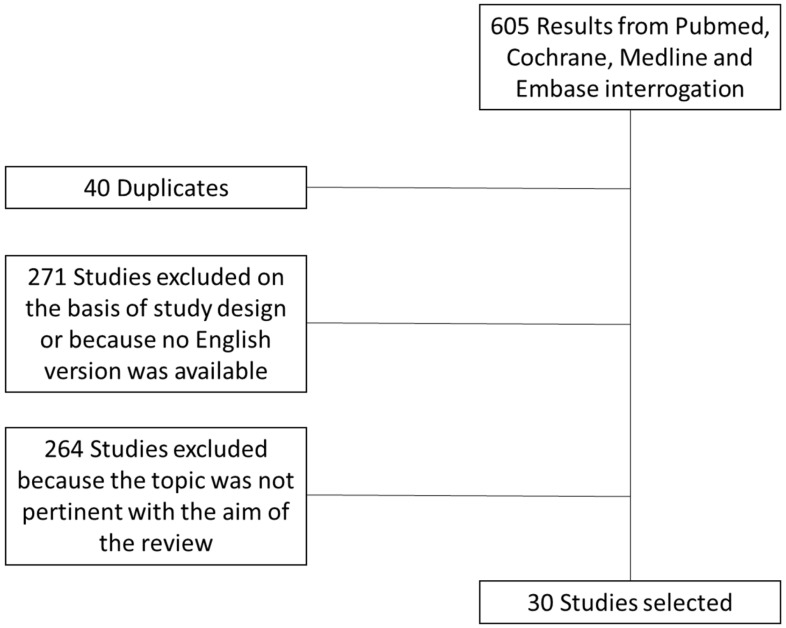
Flowchart of the study selection process. We reported the selection process of the studies included in this review.

**Table 1 nutrients-14-03908-t001:** **Summary of the pre-clinical study selected.** Abbreviations: vit D, vitamin D; SSc, systemic sclerosis; VDR, vitamin D receptors; k/o, knock out; TGF, transforming growth factor; Th, T helper; Treg, regulatory T cells; IL, interleukin; RORγ, Retinoic-acid-receptor-related orphan nuclear receptor gamma; HOCl, hypochlorous acid; MMP, metalloproteinase; TIMP, tissue inhibitor of metalloproteinase 1.

Authors	Journal and Year	Methods	Endpoint	Main Findings
Slominski et al. [27]	J. Clin. Endocrinol. Metab., 2013	Human dermal fibroblasts from SSc and healthy controls in vitro; murine models of bleomycin-induced skin fibrosis	Test the potential antifibrogenic activity of vit D analog	The noncalcemic analogs of vit D, 20(OH)D_3_, and 20,23(OH)_2_D_3_ inhibited TGF-β 1-induced collagen and hyaluronan synthesis similarly to 1,25(OH)_2_D_3_ in cultured human fibroblasts 20(OH)D_3_ suppressed fibrogenesis in bleomycin-model mice as demonstrated by skin biopsies
Usategui et al. [28]	Arch. Dermatol. Res., 2014	Bleomycin-induced fibrosis mouse model of scleroderma	Prove the potential of topical vit D to treat skin fibrosis	In topical calcipotriol-treated mice, the dermal collagen area and the dermal thickness were significantly reduced.
Zerr et al. [29]	Ann. Rheum. Dis., 2015	Fibroblasts from SSc patients and healthy controls; induced bleomycin skin fibrosis in VDR k/o mice	Role of VDR signaling in SSc fibrosis	VDR expression (mRNA and protein) is reduced in SSc fibroblasts and the murine model of skin fibrosis in a TGFbeta dependent manner. Vit D analog paricalcitol, through VDR signaling, inhibited TGF-beta signaling and ameliorated experimental fibrosis
Terao et al. [30]	Dermatoendocrinology, 2015	Normal human fibroblasts cultures; fibroblasts from a bleomycin-induced scleroderma mouse model	Effect of vit D analog on Th2 cytokine-induced periostin production by fibroblasts	Vit D analog maxacalcitol decreased the density of collagen bundles and periostin expression in the murine model; moreover, it decreased the expression of periostin in dermal fibroblasts, and the Th2 cytokine and TGFbeta-induced expression of periostin and Col1A1.
Di Liberto et al. [31]	Clin. Exp. Rheumatol., 2019	Treg isolated from blood and sera samples of SSc and controls	Effect of vit D supplementation on Treg in SSc patients	Tregs from SSc patients taking vit D increased in percentages; Tregs obtained from SSc patients failed to suppress T cell proliferation even after stimulation with vit D. However, vit D induced the production of IL-10
Janjetovic et al. [32]	Endocrinology, 2021	Bleomycin-Mouse model and murine fibroblasts	Effect of 20(OH)D_3_ on fibroblasts and role of RORγ	20(OH)D_3_ inhibited proliferation of RORγ^+/+^ fibroblasts and TGbeta-induced collagen synthesis.
Ge et al. [33]	Biochem. Biophys. Res. Commun., 2022	VDR knockout mice; HOCl-induced mice model of scleroderma	Explore the mechanism of VDR in SSc	VDR deficiency in keratinocytes promoted fibrosis; ablation of VDR in epidermidis upregulated expression of pro-inflammatory cytokines and aggravated fibrosis.
Brown Lobbins et al. [34]	Int. J. Mol. Sci., 2022	Skin biopsy from a bleomycin-induced scleroderma mouse model	Vit D-analog capacity of suppression the fibrosis in a murine model	17,20S(OH)_2_pD suppressed total collagen content, prevented the development of increased dermal thickness in a murine model, and suppressed TGF-β collagene synthesis in the murine model
Brown Lobbins et al. [35]	Int. J. Mol. Sci., 2022	Dermal cultured fibroblasts from SSc patients and controls	Vit D-analog capacity of suppression of collagen production by fibroblasts	17,20S(OH)_2_pD increased MMP-1 in dermal fibroblasts and decreased TIMP-1 protein synthesis and modulated mediators of fibrosis in vitro.

**Table 2 nutrients-14-03908-t002:** **Summary of the clinical study selected.** Abbreviations: SSc, systemic sclerosis; vit D, vitamin D; DLCO, diffusing lung capacity of the lung for carbon monoxide; ANA, antinuclear antibodies; BMD, bone mineral density; BMI, body mass index; dcSSc, diffuse cutaneous SSc; lcSSc, limited cutaneous SSc; VDR, vitamin D receptors; CT, computed tomography; DUs, digital ulcers; ET-1, endothelin 1; FGF-23, Fibroblast growth factor 23.

Authors	Journal and Year	Design of the Trial	Patients	Endpoints	Main Findings
Rios-Fernandez et al. [36]	Clin. Exp. Rheumatol., 2012	Case-control	100 SSc vs. 100 control	Prevalence of osteopenia/osteoporosis among SSc patients and controls; association of vit D levels with clinical manifestations of SSc	SSc patients had a higher prevalence of osteopenia and osteoporosis; vit D levels are associated with calcinosis, heart involvement, DLCO, and ANA positivity
Ibn Yacoub et al. [37]	Rheumatol. Int., 2012	Case-control	60 SSc patients vs. 60 age and gender-matched controls	Comparison of the BMD in women with SSc and controls; the relationship between vit D status and disease parameters and BMD	BMD was significantly lower in SSc patients than in controls; in multiple regression models, there were significant correlations between BMD and longer duration of SSc, severe joint involvement, malabsorption syndrome, and the positivity of anti-DNA topoisomerase I antibodies; Vitamin D levels were correlated with the severity of joint pain, with anti-DNA topoisomerase I positivity and with BMD in the lumbar spine and femoral neck
Atteritano et al. [38]	PloS ONE, 2013	Case-control	54 postmenopausal women with SSc and 54 postmenopausal controls	Comparison of BMD in SS patients and healthy controls; the prevalence of vertebral fractures	BMD at the lumbar spine, femoral neck, and total femur and ultrasound parameters at calcaneus were significantly lower in SSc patients, with a higher prevalence of vertebral fractures; SSc patients had a lower vit D plasma concentration, which was inversely related to BMD
Corrado et al. [39]	PloS ONE, 2015	Case-control	64 SSc vs. 35 healthy controls	Evaluations of BMD, BMI, and vit D levels in two skin subsets (limited or diffuse) of SSc patients	BMD is significantly lower in dcSSc than in lcSSc and healthy controls; Vit D serum levels are higher in healthy controls than in SSc patients; among them, those affected by dcSSc showed lower levels than those with lcSSc in dcSSc (*p* < 0.001); vit D levels are not associated with internal organ involvement
Sampaio-Barros et al. [40]	Rev. Bras. Reumatol., 2016	Cross-sectional	38 diffuse SSc patients	Correlation of vit D levels with organ involvement, antibody profile, BMD, results of questionnaires assessing the quality of life, nailfold capillaroscopy findings	Vit D levels were not correlated with organ involvement; vit d was lower in Scl-70+ subjects (*p* = 0.039); vit D levels were negatively correlated with quality of life, BMD and capillaroscopy findings
Kamal et al. [41]	Immunol. Inves., 2016	Case-control	30 SSc patients and 60 healthy subjects	Evaluation of the potential association of VDR gene polymorphisms ApaI, and TaqI with SSc susceptibility in the Egyptian population.	No significant association of VDR ApaI and TaqI polymorphisms with SSc susceptibility
Atteritano et al. [42]	Int. J. Mol. Sci., 2016	Case-control	40 SSc patients vs. 40 healthy control	Assess the prevalence of vitamin D insufficiency and correlation with clinical parameters in SSc	Lower vitamin D levels were found in SSc patients vs healthy control. Skin involvement and pulmonary hypertension were associated with vitamin D deficiency
Groseanu et al. [43]	Eur. J. Rheumatol.	Cross-sectional	51 SSc patients	Evaluation of vitamin D concentration in SSc patients and its possible association with clinical manifestations	High prevalence of hypovitaminosis D (only 9.8% of subjects reached satisfactory levels); no correlation between vitamin D concentration and autoantibody profile, the extent of skin involvement; direct correlation of vitamin D with the DLCO, diastolic dysfunction, digital contractures, and muscle weakness
Trombetta et al. [44]	PloS ONE, 2017	Retrospective cohort	154 SSc patients	Evaluation of possible correlations between vit D concentration and clinical manifestations	Vit D plasma levels were similar among patients with different clinical phenotypes and autoantibody positivity; vit. D concentrations were lower in those with bibasal fibrotic changes at lung CT scan
Giuggioli et al. [45]	Clin. Rheumatol., 2017	Cross sectional	140 SSc patients, 49 supplemented and 91 not supplemented	Evaluation of possible correlations between vit D supplementation and clinical manifestations	SSc patients undergoing vit D supplementation showed higher vit D plasma levels, a lower prevalence of autoimmune thyroiditis, and a higher frequency of anticentromere antibodies
Park et al. [46]	Clin. Rheumatol., 2017	Case-control	40 SSc women vs. 80 healthy controls	Investigate the association of vit D deficiency with digital ulcers (DUs), carotid intima-media thickness and brachial-ankle pulse wave velocity	Vit D deficiency was an independent risk factor for DUs development, while it was not associated with atherosclerosis or arterial stiffness
Zhang et al. [47]	Int. J. Rheum. Dis., 2017	Case-control	60 SSc vs. 60 healthy controls	Evaluation of vit D serum levels in SSc patients and healthy controls; evaluation of the potential association between vit. D and clinical features	Serum vit D levels were significantly lower in SSc patients, with no associations with clinical features of the disease
Ahmadi et al. [48]	Iran. J. Public Health, 2017	Case-control	60 SSc patients vs. 30 healthy controls	Comparison of serum Klotho, FGF-23, and 25-hydroxy vit D levels in the SSc patients and healthy controls.	Serum Klotho and vit D concentrations are significantly lower in SSc patients than in the control group; no significant difference in FGF-23 levels between groups
Hajialilo et al. [49]	Rheumatol. Int., 2017	Case-control	60 SSc patients vs. 60 healthy controls	Comparison of serum ET-1, α-Klotho, and vit D levels in patients with lcSSc and dcSSc scleroderma compared to healthy subjects	ET-1 was higher in SSc patients, while α-Klotho and 25(OH)D_3_ were lower in patients; Vit D levels were not associated with a specific autoantibody pattern
Kotyla et al. [50]	J. Clin. Med., 2018	Case-control	48 SSc patients vs. 23 healthy controls	Assessment of the levels of vit D, α-Klotho, and FGF23 in SSc patients; association with clinical features	Vit D levels are lower in SSc patients. Vit D was not associated with the extent of skin involvement or disease severity
Gupta et al. [51]	Indian Dermatol. Online J., 2018	Case-control	38 SSc patients vs. 38 health controls	Evaluation of vit D levels in SSc, in comparison to healthy controls and association with the extent of skin involvement	Vit D levels were lower in SSc patients and inversely associated with skin involvement assessed by the modified Rodnan skin score
Li et al. [52]	Arch. Med. Res., 2019	Case-control	100 SSc patients and 100 healthy controls	Evaluation of the potential association of eight VDR gene polymorphisms ApaI, and TaqI with SSc susceptibility	ApaI and BglI polymorphism genotypes were significantly associated with the risk of SSc.
Caimmi et al. [53]	Int. J. Rheum. Dis., 2019	Retrospective cohort	65 SSc patients	Evaluation of the association between vit D levels variation over time and development of DUs	The reduction of vit D level was correlated with an increased risk of developing DUs
Horvath et al. [54]	Arthritis Res. Ther., 2019	Case-control	44 SSc patients vs. 33 healthy controls	Evaluation of bone alterations in SSc	BMD measured at the femoral neck and lumbar spine was lower in SSc patients than in controls; hypovitaminosis D was more frequent in SSc patients (60%) than in controls (39.3%; *p* = 0.003)
Hax et al. [55]	J. Clin. Rheumatol., 2020	Case-control	50 SSc patients vs. 35 healthy controls	Evaluation of the correlation between serum levels of vit D and cytokines concentrations in SSc	Despite a more frequent vit D supplementation, SSc patients showed lower vit D levels; vit D plasma concentration was not correlated with cytokine profile
Runowska et al. [56]	Reumatologia, 2021	Retrospective cohort	112 patients with connective tissue disease; 44 with SSc	Evaluation of hypovitaminosis D prevalence among rheumatic diseases patients	Hypovitaminosis D is highly prevalent in SSc patients, despite vitamin D supplementation

## Data Availability

Not applicable.

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
