# Peer review of "Vitamin D in Systemic Sclerosis: A Review"

_nutrients, 2022, doi:10.3390/nu14193908_

Round 1
Reviewer 1 Report
The objective of the study is current and important. The selection criteria is appropriate. The results section is thorough and provides a good explanation of most of the papers selected. The last paragraph of the discussion provides detailed potential association of vitamin D levels with specific disease domains. The loose and controversial association for interstitial lung disease development is properly addressed but no definite characterization can be drawn as would be expected from a review paper.
The English language usage is appropriate
Author Response
Unfortunately, the heterogeneity of the findings available in the current literature did not allow to draw definitive conclusions about the association between vitamin D levels and ILD; however, this seems a field of investigation worth of future ad hoc studies.
Reviewer 2 Report
1. One of the major limitations of this paper is the small number of studies included (30 studies included compared to >600 studies found on this topic). Please provide more details on the rationale to exclude so many papers. Particularly when it comes to excluding papers because of study design or because the topic was not pertinent. Moreover, you included papers that have full-text availability, what does full-text availability mean? does it mean that you included open access paper only?
2. In the discussion, you stated that one of the most promising fields is ILD and digital ulcer association with low Vit D level, but in the results, you stated that this association is controversial. Please clarify. Also, both digital ulcers and ILD are seen in active disease, so is active disease lead to a low Vit D level, or is a low Vit D level a risk factor for ILD and digital ulcers? The study design of this paper is limited; thus, you were not been able to answer pertinent specific, and meaningful questions possibly because of the large exclusion of published papers.
3. No gender, age, disease duration, activity, etc. were included in the analysis. Also, are other fat-soluble vitamin levels affected by the disease?
Author Response
Thank you very much for your suggestions; please, find a point-to-point rebuttal:
1. Our search strategy led to the identification of many papers (264, also see Fig 1) which did not directly discuss the role of vit. D in SSc; moreover, in this systematic review we included only original papers with specific study designs (cohort studies, case control studies, RCT) as discussed in the methods. This approach led to the exclusion of many other papers. Also note that, when we defined full text availability, we meant the availability of the full version of the paper online, which is a different (and larger) concept than the availability of an open access full text. We have better specified this in the methods section.
Finally, the reviewer might agree with us that one of the purposes of a review paper is to propose to the reader a careful selection of the existing literature, based on rigorous criteria. We did our best to include all relevant papers; however, we will be more than happy to include others that we might have missed if suggested by the reviewer.
2. The literature about the postulated association between vit. D and DU/ILD is indeed controversial. In this revised version, we rephrased our comments to be more linear and consistent in discussing this intriguing issue. We also added a comment on the potential role of low vit D levels as a risk factor for organ involvement. We did not find other papers specifically assessing these issues; so, we do not think that study selection might have led to a bias in the identification of potential associations. However, once again, we would be happy to review any other paper(s) pointed out by the reviewer.
3. The reviewer is correct in pointing out that, in Table 1, we just presented the main findings of the selected studies. We did so in the interest of brevity, since the table is already quite busy and difficult to read.
To the best of our knowledge, no data are available in the literature about the levels of other fat soluble vitamins in SSc patients. We added a comment about this in the text